# Improved machine learning algorithm for predicting ground state properties

Laura Lewis[1,5], Hsin-Yuan Huang [1,2,6] ✉, Viet T. Tran[3], Sebastian Lehner[3], Richard Kueng[3] & John Preskill[1,4]

Finding the ground state of a quantum many-body system is a fundamental problem in quantum physics. In this work, we give a classical machine learning (ML) algorithm for predicting ground state properties with an inductive bias encoding geometric locality. The proposed ML model can efficiently predict ground state properties of an $n$-qubit gapped local Hamiltonian after learning from only $\mathcal{O}(\log(n))$ data about other Hamiltonians in the same quantum phase of matter. This improves substantially upon previous results that require $\mathcal{O}(n^c)$ data for a large constant $c$. Furthermore, the training and prediction time of the proposed ML model scale as $\mathcal{O}(n \log n)$ in the number of qubits $n$. Numerical experiments on physical systems with up to 45 qubits confirm the favorable scaling in predicting ground state properties using a small training dataset.

Finding the ground state of a quantum many-body system is a fundamental problem with far-reaching consequences for physics, materials science, and chemistry. Many powerful methods[1-7] have been proposed, but classical computers still struggle to solve many general classes of the ground state problem. To extend the reach of classical computers, classical machine learning (ML) methods have recently been adapted to study this and related problems both empirically and theoretically[8-35]. A recent work[36] proposes a polynomial-time classical ML algorithm that can efficiently predict ground state properties of gapped geometrically local Hamiltonians, after learning from data obtained by measuring other Hamiltonians in the same quantum phase of matter. Furthermore[36], shows that under a widely accepted conjecture, no polynomial-time classical algorithm can achieve the same performance guarantee. However, although the ML algorithm given in[36] uses a polynomial amount of training data and computational time, the polynomial scaling $\mathcal{O}(n^c)$ has a very large degree $c$. Here, $f(x) = \mathcal{O}(g(x))$ denotes that $f(x)$ is asymptotically upper bounded by $g(x)$ up to constant factors with respect to the limit $n \to \infty$. Moreover, when the prediction error $\epsilon$ is small, the amount of training data grows exponentially in $1/\epsilon$, indicating that a very small prediction error cannot be achieved efficiently.

In this work, we present an improved ML algorithm for predicting ground state properties. We consider an $m$-dimensional vector $x \in [-1, 1]^m$ that parameterizes an $n$-qubit gapped geometrically local Hamiltonian given as

$$H(x) = \sum_j h_j(\vec{x}_j), \qquad (1)$$

where $x$ is the concatenation of constant-dimensional vectors $\vec{x}_1, \ldots, \vec{x}_L$ parameterizing the few-body interaction $h_j(\vec{x}_j)$. Let $\rho(x)$ be the ground state of $H(x)$ and $O$ be a sum of geometrically local observables with $\|O\|_\infty \le 1$. We assume that the geometry of the $n$-qubit system is known, but we do not know how $h_j(\vec{x}_j)$ is parameterized or what the observable $O$ is. The goal is to learn a function $h^*(x)$ that approximates the ground state property $\mathrm{Tr}(O\rho(x))$ from a classical dataset,

$$(x_\ell, y_\ell), \quad \forall \ell = 1, \ldots, N, \qquad (2)$$

where $y_\ell \approx \mathrm{Tr}(O\rho(x_\ell))$ records the ground state property for $x_\ell \in [-1, 1]^m$ sampled from an arbitrary unknown distribution $\mathcal{D}$. Here,

[1]California Institute of Technology, Pasadena, CA, USA. [2]Massachusetts Institute of Technology, Cambridge, MA, USA. [3]Johannes Kepler University, Linz, Austria. [4]AWS Center for Quantum Computing, Pasadena, CA, USA. [5]Present address: University of Cambridge, Cambridge, UK. [6]Present address: Google Quantum AI, Venice, CA, USA. ✉e-mail: hsinyuan@caltech.edu

$y_\ell \approx \mathrm{Tr}(O\rho(x_\ell))$ means that $y_\ell$ has additive error at most $\epsilon$. If $y_\ell = \mathrm{Tr}(O\rho(x_\ell))$, the rigorous guarantees improves.

The setting considered in this work is very similar to that in[36], but we assume the geometry of the $n$-qubit system to be known, which is necessary to overcome the sample complexity lower bound of $N = n^{\Omega(1/\epsilon)}$ given in[36]. Here, $f(x) = \Omega(g(x))$ denotes that $f(x)$ is asymptotically lower bounded by $g(x)$ up to constant factors. One may compare the setting to that of finding ground states using adiabatic quantum computation[37–44]. To find the ground state property $\mathrm{Tr}(O\rho(x))$ of $H(x)$, this class of quantum algorithms requires the ground state $\rho_0$ of another Hamiltonian $H_0$ stored in quantum memory, explicit knowledge of a gapped path connecting $H_0$ and $H(x)$, and an explicit description of $O$. In contrast, here we focus on ML algorithms that are entirely classical, have no access to quantum state data, and have no knowledge about the Hamiltonian $H(x)$, the observable $O$, or the gapped paths between $H(x)$ and other Hamiltonians.

The proposed ML algorithm uses a nonlinear feature map $x \mapsto \phi(x)$ with a geometric inductive bias built into the mapping. At a high level, the high-dimensional vector $\phi(x)$ contains nonlinear functions for each geometrically local subset of coordinates in the $m$-dimensional vector $x$. Here, the geometry over coordinates of the vector $x$ is defined using the geometry of the $n$-qubit system. The ML algorithm learns a function $h^*(x) = \mathbf{w}^* \cdot \phi(x)$ by training an $\ell_1$-regularized regression (LASSO)[45–47] in the feature space. An overview of the ML algorithm is shown in Fig. 1. We prove that given $\epsilon = \Theta(1)$, Here, the notation $f(x) = \Theta(g(x))$ denotes that $f(x) = \mathcal{O}(g(x))$ and $f(x) = \Omega(g(x))$ both hold. Hence, $f(x)$ is asymptotically equal to $g(x)$ up to constant factors. the improved ML algorithm can use a dataset size of

$$N = \mathcal{O}(\log(n)), \tag{3}$$

to learn a function $h^*(x)$ with an average prediction error of at most $\epsilon$,

$$\mathop{\mathbb{E}}_{x \sim \mathcal{D}} |h^*(x) - \mathrm{Tr}(O\rho(x))|^2 \le \epsilon, \tag{4}$$

with high success probability.

The sample complexity $N = \mathcal{O}(\log(n))$ of the proposed ML algorithm improves substantially over the sample complexity of $N = \mathcal{O}(n^c)$ in the previously best-known classical ML algorithm[36], where $c$ is a very large constant. The computational time of both the improved ML algorithm and the ML algorithm in[36] is $\mathcal{O}(nN)$. Hence, the logarithmic sample complexity $N$ immediately implies a nearly linear computational time. In addition to the reduced sample complexity and computational time, the proposed ML algorithm works for any distribution over $x$, while the best previously known algorithm[36] works only for the uniform distribution over $[-1, 1]^m$. Furthermore, when we consider the scaling with the prediction error $\epsilon$, the best known classical ML algorithm in[36] has a sample complexity of $N = n^{\mathcal{O}(1/\epsilon)}$, which is exponential in $1/\epsilon$. In contrast, the improved ML algorithm has a sample complexity of $N = \log(n) 2^{\mathrm{polylog}(1/\epsilon)}$, which is quasi-polynomial in $1/\epsilon$.

We also discuss a generalization of the proposed ML algorithm to predicting ground state representations when trained on classical shadow representations[48–52]. In this setting, the proposed ML algorithm yields the same reduction in sample and time complexity compared to[36] for predicting ground state representations.

## Results

The central component of the improved ML algorithm is the geometric inductive bias built into our feature mapping $x \in [-1,1]^m \mapsto \phi(x) \in \mathbb{R}^{m_\phi}$. To describe the ML algorithm, we first need to present some definitions relating to this geometric structure.

### Definitions of the geometric inductive bias

We consider $n$ qubits arranged at locations, or sites, in a $d$-dimensional space, e.g., a spin chain ($d = 1$), a square lattice ($d = 2$), or a cubic lattice ($d = 3$). This geometry is characterized by the distance $d_{\mathrm{qubit}}(i, i')$ between any two qubits $i$ and $i'$. Using the distance $d_{\mathrm{qubit}}$ between qubits, we can define the geometry of local observables. Given any two observables $O_A, O_B$ on the $n$-qubit system, we define the distance $d_{\mathrm{obs}}(O_A, O_B)$ between the two observables as the minimum distance between the qubits that $O_A$ and $O_B$ act on. We also say an observable is geometrically local if it acts nontrivially only on nearby qubits under the distance metric $d_{\mathrm{qubit}}$. We then define $S^{(\mathrm{geo})}$ as the set of all geometrically local Pauli observables, i.e., geometrically local observables that belong to the set $\{I, X, Y, Z\}^{\otimes n}$. The size of $S^{(\mathrm{geo})}$ is $\mathcal{O}(n)$, linear in the total number of qubits.

With these basic definitions in place, we now define a few more geometric objects. The first object is the set of coordinates in the $m$-dimensional vector $x$ that are close to a geometrically local Pauli observable $P$. This is formally given by,

$$I_P \triangleq \left\{ c \in \{1, \ldots, m\} : d_{\mathrm{obs}}(h_{j(c)}, P) \le \delta_1 \right\}, \tag{5}$$

where $h_{j(c)}$ is the few-body interaction term in the $n$-qubit Hamiltonian $H(x)$ whose parameters $\vec{x}_{j(c)}$ include the variable $x_c \in [-1, 1]$, and $\delta_1$ is an efficiently computable hyperparameter that is determined later. Each variable $x_c$ in the $m$-dimensional vector $x$ corresponds to exactly one interaction terms $h_{j(c)} = h_{j(c)}(\vec{x}_{j(c)})$, where the parameter vector $\vec{x}_{j(c)}$ contains the variable $x_c$. Intuitively, $I_P$ is the set of coordinates that have the strongest influence on the function $\mathrm{Tr}(P\rho(x))$.

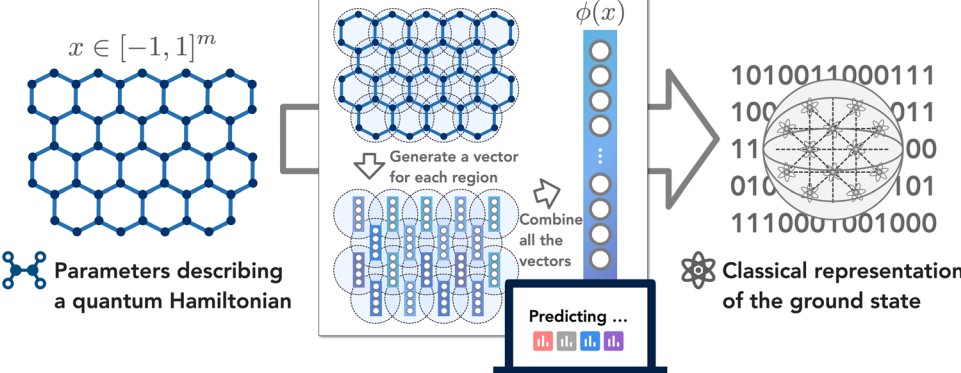

**Fig. 1 | Overview of the proposed machine learning algorithm.** Given a vector $x \in [-1, 1]^m$ that parameterizes a quantum many-body Hamiltonian $H(x)$, the algorithm uses a geometric structure to create a high-dimensional vector $\phi(x) \in \mathbb{R}^{m_\phi}$. The ML algorithm then predicts properties or a representation of the ground state $\rho(x)$ of Hamiltonian $H(x)$ using the $m_\phi$-dimensional vector $\phi(x)$.

The second geometric object is a discrete lattice over the space $[-1, 1]^m$ associated to each subset $I_P$ of coordinates. For any geometrically local Pauli observable $P \in S^{(\text{geo})}$, we define $X_P$ to contain all vectors $x$ that take on value 0 for coordinates outside $I_P$ and take on a set of discrete values for coordinates inside $I_P$. Formally, this is given by

$$X_P \triangleq \left\{ \begin{array}{l} x \in [-1,1]^m : \text{if } c \notin I_P, \, x_c = 0 \\ \text{if } c \in I_P, \, x_c \in \{0, \pm\delta_2, \pm2\delta_2, \dots, \pm1\} \end{array} \right\}, \quad (6)$$

where $\delta_2$ is an efficiently computable hyperparameter to be determined later. The definition of $X_P$ is meant to enumerate all sufficiently different vectors for coordinates in the subset $I_P \subseteq \{1, \dots, m\}$.

Now given a geometrically local Pauli observable $P$ and a vector $x$ in the discrete lattice $X_P \subseteq [-1, 1]^m$, the third object is a set $T_{x,P}$ of vectors in $[-1, 1]^m$ that are close to $x$ for coordinates in $I_P$. This is formally defined as,

$$T_{x,P} \triangleq \left\{ x' \in [-1,1]^m : -\frac{\delta_2}{2} < x_c - x'_c \leq \frac{\delta_2}{2}, \forall c \in I_P \right\}. \quad (7)$$

The set $T_{x,P}$ is defined as a thickened affine subspace close to the vector $x$ for coordinates in $I_P$. If a vector $x'$ is in $T_{x,P}$, then $x'$ is close to $x$ for all coordinates in $I_P$, but $x'$ may be far away from $x$ for coordinates outside of $I_P$. Examples of these definitions are given in Supplementary Figs. 1 and 2.

## Feature mapping and ML model

We can now define the feature map $\phi$ taking an $m$-dimensional vector $x$ to an $m_\phi$-dimensional vector $\phi(x)$ using the thickened affine subspaces $T_{x',P}$ for every geometrically local Pauli observable $P \in S^{(\text{geo})}$ and every vector $x'$ in the discrete lattice $X_P$. The dimension of the vector $\phi(x)$ is given by $m_\phi = \sum_{P \in S^{(\text{geo})}} |X_P|$. Each coordinate of the vector $\phi(x)$ is indexed by $x' \in X_P$ and $P \in S^{(\text{geo})}$ with

$$\phi(x)_{x',P} \triangleq \mathbb{1}\left[x \in T_{x',P}\right], \quad (8)$$

which is the indicator function checking if $x$ belongs to the thickened affine subspace. Recall that this means each coordinate of the $m_\phi$-dimensional vector $\phi(x)$ checks if $x$ is close to a point $x'$ on a discrete lattice $X_P$ for the subset $I_P$ of coordinates close to a geometrically local Pauli observable $P$.

The classical ML model we consider is an $\ell_1$-regularized regression (LASSO) over the $\phi(x)$ space. More precisely, given an efficiently computable hyperparameter $B > 0$, the classical ML model finds an $m_\phi$-dimensional vector $\mathbf{w}^*$ from the following optimization problem,

$$\min_{\mathbf{w} \in \mathbb{R}^{m_\phi}} \frac{1}{N} \sum_{\ell=1}^{N} |\mathbf{w} \cdot \phi(x_\ell) - y_\ell|^2, \quad (9)$$
$$\| \mathbf{w} \|_1 \leq B$$

where $\{(x_\ell, y_\ell)\}_{\ell=1}^{N}$ is the training data. Here, $x_\ell \in [-1, 1]^m$ is an $m$-dimensional vector that parameterizes a Hamiltonian $H(x)$ and $y_\ell$ approximates $\text{Tr}(O\rho(x_\ell))$. The learned function is given by $h^*(x) = \mathbf{w}^* \cdot \phi(x)$. The optimization does not have to be solved exactly. We only need to find a $\mathbf{w}^*$ whose function value is $\mathcal{O}(\epsilon)$ larger than the minimum function value. There is an extensive literature[53–59] improving the computational time for the above optimization problem. The best known classical algorithm[58] has a computational time scaling linearly in $m_\phi / \epsilon^2$ up to a log factor, while the best known quantum algorithm[59] has a computational time scaling linearly in $\sqrt{m_\phi} / \epsilon^2$ up to a log factor.

## Rigorous guarantee

The classical ML algorithm given above yields the following sample and computational complexity. This theorem improves substantially upon the result in[36], which requires $N = n^{\mathcal{O}(1/\epsilon)}$. The proof idea is given in Section "Methods", and the detailed proof is given in Supplementary Sections 1, 2, 3. Using the proof techniques presented in this work, one can show that the sample complexity $N = \log(n/\delta)2^{\text{polylog}(1/\epsilon)}$ also applies to any sum of few-body observables $O = \sum_j O_j$ with $\sum_j \|O_j\|_\infty \leq 1$, even if the operators $\{O_j\}$ are not geometrically local.

**Theorem 1.** (Sample and computational complexity). Given $n, \delta > 0, \frac{1}{e} > \epsilon > 0$ and a training data set $\{x_\ell, y_\ell\}_{\ell=1}^{N}$ of size

$$N = \log(n/\delta)2^{\text{polylog}(1/\epsilon)}, \quad (10)$$

where $x_\ell$ is sampled from an unknown distribution $\mathcal{D}$ and $|y_\ell - \text{Tr}(O\rho(x_\ell))| \leq \epsilon$ for any observable $O$ with eigenvalues between $-1$ and $1$ that can be written as a sum of geometrically local observables. With a proper choice of the efficiently computable hyperparameters $\delta_1, \delta_2$, and $B$, the learned function $h^*(x) = \mathbf{w}^* \cdot \phi(x)$ satisfies

$$\mathbb{E}_{x \sim \mathcal{D}} |h^*(x) - \text{Tr}(O\rho(x))|^2 \leq \epsilon \quad (11)$$

with probability at least $1 - \delta$. The training and prediction time of the classical ML model are bounded by $\mathcal{O}(nN) = n \log(n/\delta)2^{\text{polylog}(1/\epsilon)}$.

The output $y_\ell$ in the training data can be obtained by measuring $\text{Tr}(O\rho(x_\ell))$ for the same observable $O$ multiple times and averaging the outcomes. Alternatively, we can use the classical shadow formalism[48–52,60] that performs randomized Pauli measurements on $\rho(x_\ell)$ to predict $\text{Tr}(O\rho(x_\ell))$ for a wide range of observables $O$. We can also combine Theorem 1 and the classical shadow formalism to use our ML algorithm to predict ground state representations, as seen in the following corollary. This allows one to predict ground state properties $\text{Tr}(O\rho(x))$ for a large number of observables $O$ rather than just a single one. We present the proof of Corollary 1 in Supplementary Section 3B.

**Corollary 1.** Given $n, \delta > 0, \frac{1}{e} > \epsilon > 0$ and a training data set $\{x_\ell, \sigma_T(\rho(x_\ell))\}_{\ell=1}^{N}$ of size

$$N = \log(n/\delta)2^{\text{polylog}(1/\epsilon)}, \quad (12)$$

where $x_\ell$ is sampled from an unknown distribution $\mathcal{D}$ and $\sigma_T(\rho(x_\ell))$ is the classical shadow representation of the ground state $\rho(x_\ell)$ using $T$ randomized Pauli measurements. For $T = \tilde{\mathcal{O}}(\log(n)/\epsilon^2)$, then the proposed ML algorithm can learn a ground state representation $\hat{\rho}_{N,T}(x)$ that achieves

$$\mathbb{E}_{x \sim \mathcal{D}} |\text{Tr}(O\hat{\rho}_{N,T}(x)) - \text{Tr}(O\rho(x))|^2 \leq \epsilon \quad (13)$$

for any observable $O$ with eigenvalues between $-1$ and $1$ that can be written as a sum of geometrically local observables with probability at least $1 - \delta$.

We can also show that the problem of estimating ground state properties for the class of parameterized Hamiltonians $H(x) = \sum_j h_j(\vec{x}_j)$ considered in this work is hard for non-ML algorithms that cannot learn from data, assuming the widely believed conjecture that NP-complete problems cannot be solved in randomized polynomial time. This is a manifestation of the computational power of data studied in[61]. The proof of Proposition 1 in[36] constructs a parameterized Hamiltonian $H(x)$ that belongs to the family of parameterized Hamiltonians considered in this work and hence establishes the following.

**Proposition 1.** (A variant of Proposition 1 in[36]). Consider a randomized polynomial-time classical algorithm $\mathcal{A}$ that does not learn from data. Suppose for any smooth family of gapped 2D Hamiltonians $H(x) = \sum_j h_j(\vec{x}_j)$ and any single-qubit observable $O, \mathcal{A}$ can compute ground state properties $\text{Tr}(O\rho(x))$ up to a constant error averaged over

$x \in [-1, 1]^m$ uniformly. Then, NP-complete problems can be solved in randomized polynomial time.

This proposition states that even under the restricted settings of considering only 2D Hamiltonians and single-qubit observables, predicting ground state properties is a hard problem for non-ML algorithms. When one consider higher-dimensional Hamiltonians and multi-qubit observables, the problem only becomes harder because one can embed low-dimensional Hamiltonians in higher-dimensional spaces.

## Numerical experiments

We present numerical experiments to assess the performance of the classical ML algorithm in practice. The results illustrate the improvement of the algorithm presented in this work compared to those considered in[36], the mild dependence of the sample complexity on the system size $n$, and the inherent geometry exploited by the ML models. We consider the classical ML models previously described, utilizing a random Fourier feature map[62]. While the indicator function feature map was a useful tool to obtain our rigorous guarantees, random Fourier features are more robust and commonly used in practice. Moreover, we still expect our rigorous guarantees to hold with this change because Fourier features can approximate any function, which is the central property of the indicator functions used in our proofs. Furthermore, we determine the optimal hyperparameters using cross-validation to minimize the root-mean-square error (RMSE) and then evaluate the performance of the chosen ML model using a test set. The models and hyperparameters are further detailed in Supplementary Section 4.

For these experiments, we consider the two-dimensional antiferromagnetic random Heisenberg model consisting of $4 \times 5 = 20$ to $9 \times 5 = 45$ spins as considered in previous work[36]. In this setting, the spins are placed on sites in a 2D lattice. The Hamiltonian is

$$H = \sum_{\langle ij \rangle} J_{ij}(X_i X_j + Y_i Y_j + Z_i Z_j), \tag{14}$$

where the summation ranges over all pairs $\langle ij \rangle$ of neighboring sites on the lattice and the couplings $\{J_{ij}\}$ are sampled uniformly from the interval [0, 2]. Here, the vector $x$ is a list of all couplings $J_{ij}$ so that the dimension of the parameter space is $m = O(n)$, where $n$ is the system size. The nonnegative interval [0, 2] corresponds to antiferromagnetic interactions. To minimize the Heisenberg interaction terms, nearby qubits have to form singlet states. While the square lattice is bipartite and lacks the standard geometric frustration, the presence of disorder makes the ground state calculation more challenging as neighboring qubits will compete in the formation of singlets due to the monogamy of entanglement[63].

We trained a classical ML model using randomly chosen values of the parameter vector $x = \{J_{ij}\}$. For each parameter vector of random couplings sampled uniformly from [0, 2], we approximated the ground state using the same method as in[36], namely with the density-matrix renormalization group (DMRG)[64] based on matrix product states (MPS)[65]. The classical ML model was trained on a data set $\{x_\ell, \sigma_T(\rho(x_\ell))\}_{\ell=1}^N$ with $N$ randomly chosen vectors $x$, where each $x$ corresponds to a classical representation $\sigma_T(\rho(x_\ell))$ created from $T$ randomized Pauli measurements[48]. For a given training set size $N$, we conduct 4-fold cross validation on the $N$ data points to select the best hyperparameters, train a model with the best hyperparameters on the $N$ data points, and test the performance on a test set of size $N$. Further details are discussed in Supplementary Section 4.

The ML algorithm predicted the classical representation of the ground state for a new vector $x$. These predicted classical representations were used to estimate two-body correlation functions, i.e., the expectation value of

$$C_{ij} = \frac{1}{3}(X_i X_j + Y_i Y_j + Z_i Z_j), \tag{15}$$

for each pair of qubits $\langle ij \rangle$ on the lattice. Here, we are using the combination of our ML algorithm with the classical shadow formalism as described in Corollary 1, leveraging this more powerful technique to predict a large number of ground state properties.

In Fig. 2A, we can clearly see that the ML algorithm proposed in this work consistently outperforms the ML models implemented in[36], which includes the rigorous polynomial-time learning algorithm based on Dirichlet kernel proposed in[36], Gaussian kernel regression[66,67], and infinite-width neural networks[68,69]. Figure 2A (Left) and (Center) show that as the number $T$ of measurements per data point or the training set size $N$ increases, the prediction performance of the proposed ML algorithm improves faster than the other ML algorithms. This observation reflects the improvement in the sample complexity dependence on prediction error $\epsilon$. The sample complexity in[36] depends exponentially on $1/\epsilon$, but Theorem 1 establishes a quasi-polynomial dependence on $1/\epsilon$. From Fig. 2A (Right), we can see that the ML algorithms do not yield a substantially worse prediction error as the system size $n$ increases. This observation matches with the $\log(n)$ sample complexity in Theorem 1, but not with the poly($n$) sample complexity proven in[36]. These improvements are also relevant when comparing the ML predictions to actual correlation function values. Figure 3 in[36] illustrates that for the average prediction error achieved in their work, the predictions by the ML algorithm match the simulated values closely. In this work, we emphasize that significantly less training data is needed to achieve the same prediction error[36] and agree with the simulated values.

An important step for establishing the improved sample complexity in Theorem 1 is that a property on a local region $R$ of the quantum system only depends on parameters in the neighborhood of region $R$. In Fig. 2B, we visualize where the trained ML model is focusing on when predicting the correlation function over a pair of qubits. A thicker and darker edge is considered to be more important by the trained ML model. Each edge of the 2D lattice corresponds to a coupling $J_{ij}$. For each edge, we sum the absolute values of the coefficients in the ML model that correspond to a feature that depends on the coupling $J_{ij}$. We can see that the ML model learns to focus only on the neighborhood of a local region $R$ when predicting the ground state property.

## Discussion

The classical ML algorithm and the advantage over non-ML algorithms as proven in[36] illustrate the potential of using ML algorithms to solve challenging quantum many-body problems. However, the classical ML model given in[36] requires a large amount of training data. Although the need for a large dataset is a common trait in contemporary ML algorithms[70–72], one would have to perform an equally large number of physical experiments to obtain such data. This makes the advantage of ML over non-ML algorithms challenging to realize in practice. The sample complexity $N = \mathcal{O}(\log n)$ of the ML algorithm proposed here illustrates that this advantage could potentially be realized after training with data from a small number of physical experiments. The existence of a theoretically backed ML algorithm with a $\log(n)$ sample complexity raises the hope of designing good ML algorithms to address practical problems in quantum physics, chemistry, and materials science by learning from the relatively small amount of data that we can gather from real-world experiments.

Despite the progress in this work, many questions remain to be answered. Recently, powerful machine learning models such as graph neural networks have been used to empirically demonstrate a favorable sample complexity when leveraging the local structure of Hamiltonians in the 2D random Heisenberg model[29,30]. Is it possible to obtain rigorous theoretical guarantees for the sample complexity of neural-network-based ML algorithms for predicting ground state properties? An alternative direction is to notice that the current results have an exponential scaling in the inverse of the spectral gap. Is the

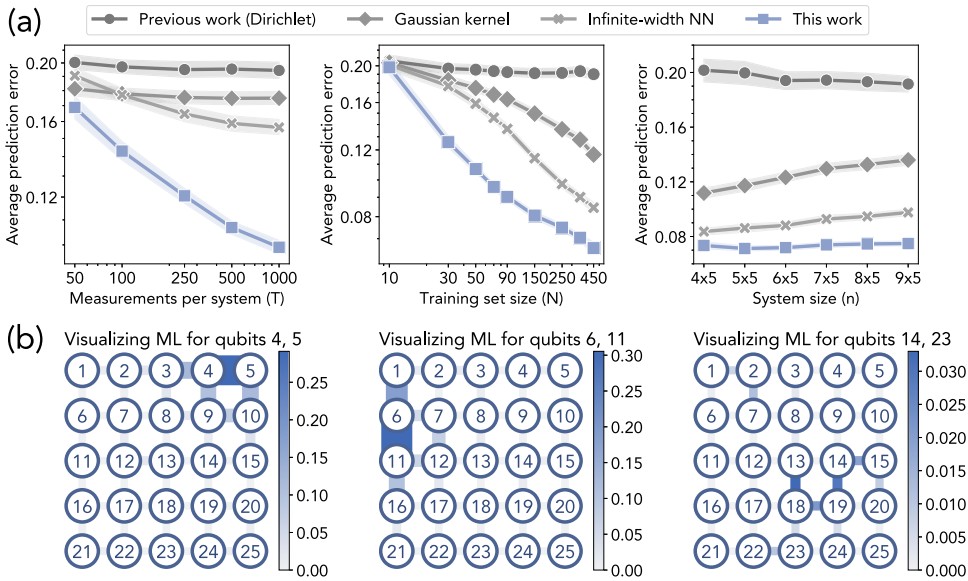

**Fig. 2 | Predicting ground state properties in 2D antiferromagnetic random Heisenberg models. a** Prediction error. Each point indicates the root-mean-square error for predicting the correlation function in the ground state (averaged over Heisenberg model instances and each pair of neighboring spins). We present log-log plots for the scaling of prediction error $\epsilon$ with $T$ and $N$: the slope corresponds to the exponent of the polynomial function $\epsilon(T)$, $\epsilon(N)$. The shaded regions show the standard deviation over different spin pairs. **b** Visualization. We plot how much each coupling $J_{ij}$ contributes to the prediction of the correlation function over different pairs of qubits in the trained ML model. Thicker and darker edges correspond to higher contributions. We see that the ML model learns to utilize the local geometric structure.

exponential scaling a fundamental nature of this problem? Or do there exist more efficient ML models that can efficiently predict ground state properties for gapless Hamiltonians?

We have focused on the task of predicting local observables in the ground state, but many other physical properties are also of high interest. Can ML models predict low-energy excited state properties? Could we achieve a sample complexity of $N = \mathcal{O}(\log n)$ for predicting any observable $O$? Another important question is whether there is a provable quantum advantage in predicting ground state properties. Could we design quantum ML algorithms that can predict ground state properties by learning from far fewer experiments than any classical ML algorithm? Perhaps this could be shown by combining ideas from adiabatic quantum computation[37–44] and recent techniques for proving quantum advantages in learning from experiments[73–77]. It remains to be seen if quantum computers could provide an unconditional super-polynomial advantage over classical computers in predicting ground state properties.

## Methods

We describe the key ideas behind the proof of Theorem 1. The proof is separated into three parts. The first part in Supplementary Section 1 describes the existence of a simple functional form that approximates the ground state property $\mathrm{Tr}(O\rho(x))$. The second part in Supplementary Section 2 gives a new bound for the $\ell_1$-norm of the Pauli coefficients of the observable $O$ when written in the Pauli basis. The third part in Supplementary Section 3 combines the first two parts, using standard tools from learning theory to establish the sample complexity corresponding to the prediction error bound given in Theorem 1. In the following, we discuss these three parts in detail.

### Simple form for ground state property

Using the spectral flow formalism[78–80], we first show that the ground state property can be approximated by a sum of local functions. First, we write $O$ in the Pauli basis as $O = \sum_{P \in \{I,X,Y,Z\}^{\otimes n}} \alpha_P P$. Then, we show that for every geometrically local Pauli observable

$P$, we can construct a function $f_P(x)$ that depends only on coordinates in the subset $I_P$ of coordinates that parameterizes interaction terms $h_j$ near the Pauli observable $P$. The function $f_P(x)$ is given by

$$f_P(x) = \alpha_P \mathrm{Tr}(P\rho(\chi_P(x))), \tag{16}$$

where $\chi_P(x) \in [-1, 1]^m$ is defined as $\chi_P(x)_c = x_c$ for coordinate $c \in I_P$ and $\chi_P(x)_c = 0$ for coordinates $c \notin I_P$. The sum of these local functions $f_P$ can be used to approximate the ground state property,

$$\mathrm{Tr}(O\rho(x)) \approx \sum_{P \in S^{(\mathrm{geo})}} f_P(x). \tag{17}$$

The approximation only incurs an $\mathcal{O}(\epsilon)$ error if we consider $\delta_1 = \Theta(\log^2(1/\epsilon))$ in the definition of $I_P$. The key point is that correlations decay exponentially with distance in the ground state of a gapped local Hamiltonian; therefore, the properties of the ground state in a localized region are not sensitive to the details of the Hamiltonian at points far from that localized region. Furthermore, the local function $f_P$ is smooth. The smoothness property allows us to approximate each local function $f_P$ by a simple discretization,

$$f_P(x) \approx \sum_{x' \in X_P} f_P(x') \mathbb{1}\left[x \in T_{x',P}\right]. \tag{18}$$

One could also use other approximations for this step, such as Fourier approximation or polynomial approximation. In fact, we apply a Fourier approximation instead in the numerical experiments, as discussed in Supplementary Section 4. For simplicity of the proof, we consider a discretization-based approximation with $\delta_2 = \Theta(1/\epsilon)$ in the definition of $T_{x',P}$ to incur at most an $\mathcal{O}(\epsilon)$ error. The point is that, for a sufficiently smooth function $f_P(x)$ that depends only on coordinates in $I_P$ and a sufficiently fine lattice over the coordinates in $I_P$, replacing $x$ by the nearest lattice point (based only on coordinates in $I_P$) causes only a small error. Using the definition of the feature map $\phi(x)$ in Eq. (8), we

have

$$\text{Tr}(O\rho(x)) \approx \sum_{P \in S^{(\text{geo})}} \sum_{x' \in X_P} f_P(x')\phi(x)_{x',P} = \mathbf{w}' \cdot \phi(x), \tag{19}$$

where $\mathbf{w}'$ is an $m_\phi$-dimensional vector indexed by $x' \in X_P$ and $P \in S^{\text{geo}}$ given by $\mathbf{w}'_{x',P} = f_P(x')$. The approximation is accurate if we consider $\delta_1 = \Theta(\log^2(1/\epsilon))$ and $\delta_2 = \Theta(1/\epsilon)$. Thus, we can see that the ML algorithm with the proposed feature mapping indeed has the capacity to approximately represent the target function $\text{Tr}(O\rho(x))$. As a result, we have the following lemma.

**Lemma 1.** (Training error bound). The function given by $\mathbf{w}' \cdot \phi(x)$ achieves a small training error:

$$\frac{1}{N}\sum_{\ell=1}^{N}|\mathbf{w}' \cdot \phi(x_\ell) - y_\ell|^2 \le 0.53\epsilon. \tag{20}$$

This lemma follows from the two facts that $\mathbf{w}' \cdot \phi(x) \approx \text{Tr}(O\rho(x))$ and $\text{Tr}(O\rho(x_\ell)) \approx y_\ell$.

### Norm inequality for observables

The efficiency of an $\ell_1$-regularized regression depends greatly on the $\ell_1$ norm of the vector $\mathbf{w}'$. Moreover, the $\ell_1$-norm of $\mathbf{w}'$ is closely related to the observable $O = \sum_j O_j$ given as a sum of geometrically local observables with $\|O\|_\infty \le 1$. In particular, again writing $O$ in the Pauli basis as $O = \sum_{Q \in \{I,X,Y,Z\}^{\otimes n}} \alpha_Q Q$, the $\ell_1$-norm $\|\mathbf{w}'\|_1$ is closely related to $\sum_Q |\alpha_Q|$, which we refer to as the Pauli 1-norm of the observable $O$. While it is well known that

$$\sum_Q |\alpha_Q|^2 = \text{Tr}(O^2)/2^n \le \|O\|_\infty^2, \tag{21}$$

there do not seem to be many known results characterizing $\sum_Q |\alpha_Q|$. To understand the Pauli 1-norm, we prove the following theorem.

**Theorem 2.** (Pauli 1-norm bound). Let $O = \sum_{Q \in \{I,X,Y,Z\}^{\otimes n}} \alpha_Q Q$ be an observable that can be written as a sum of geometrically local observables. We have,

$$\sum_Q |\alpha_Q| \le C \|O\|_\infty, \tag{22}$$

for some constant $C$.

A series of related norm inequalities are also established in[81]. However, the techniques used in this work differ significantly from those in[81].

### Prediction error bound for the ML algorithm

Using the construction of the local function $f_P(x_c, c \in I_P)$ given in Eq. (16) and the vector $\mathbf{w}'$ defined in Eq. (19), we can show that

$$\|\mathbf{w}'\|_1 \le \max_{P \in S^{(\text{geo})}}|X_P|\left(\sum_Q |\alpha_Q|\right) \le \left(1 + \frac{2}{\delta_2}\right)^{\text{poly}(\delta_1)}\left(\sum_Q |\alpha_Q|\right). \tag{23}$$

The second inequality follows by bounding the size of our discrete subset $X_P$ and noticing that $|I_P| = \text{poly}(\delta_1)$. The norm inequality in Theorem 2 then implies

$$\|\mathbf{w}'\|_1 \le C \|O\|_\infty \left(1 + \frac{2}{\delta_2}\right)^{\text{poly}(\delta_1)} \le 2^{\text{poly}\log(1/\epsilon)}, \tag{24}$$

because $\|O\|_\infty \le 1$ and $\delta_1 = \Theta(\log^2(1/\epsilon)), \delta_2 = \Theta(1/\epsilon)$. This shows that there exists a vector $\mathbf{w}'$ that has a bounded $\ell_1$-norm and achieves a small training error. The existence of $\mathbf{w}'$ guarantees that the vector $\mathbf{w}^*$

found by the optimization problem with the hyperparameter $B \ge \|\mathbf{w}'\|_1$ will yield an even smaller training error. Using the norm bound on $\mathbf{w}'$, we can choose the hyperparameter $B$ to be $B = 2^{\text{poly}\log(1/\epsilon)}$. Using standard learning theory[46,47], we can thus obtain

$$\mathbb{E}_{x \sim \mathcal{D}}|h^*(x) - \text{Tr}(O\rho(x))|^2 \le \frac{1}{N}\sum_{\ell=1}^{N}|\mathbf{w}^* \cdot \phi(x_\ell) - y_\ell|^2 + \mathcal{O}\left(B\sqrt{\frac{\log(m_\phi/\delta)}{N}}\right) \tag{25}$$

with probability at least $1 - \delta$. The first term is the training error for $\mathbf{w}^*$, which is smaller than the training error of $0.53\epsilon$ for $\mathbf{w}'$ from Lemma 1. Thus, the first term is bounded by $0.53\epsilon$. The second term is determined by $B$ and $m_\phi$, where we know that $m_\phi \le |S^{(\text{geo})}|(1 + \frac{2}{\delta_2})^{\text{poly}(\delta_1)}$ and $|S^{(\text{geo})}| = \mathcal{O}(n)$. Hence, with a training data size of

$$N = \mathcal{O}\left(\log(n/\delta)2^{\text{poly}\log(1/\epsilon)}\right), \tag{26}$$

we can achieve a prediction error of $\epsilon$ with probability at least $1 - \delta$ for any distribution $\mathcal{D}$ over $[-1, 1]^m$.

## Data availability

Source data are available for this paper. All data can be found or generated using the source code at https://github.com/lllewis234/improved-ml-algorithm[83].

## Code availability

Source code for an efficient implementation of the proposed procedure is available at https://github.com/lllewis234/improved-ml-algorithm[83].

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

## Acknowledgements

The authors thank Chi-Fang Chen, Sitan Chen, Johannes Jakob Meyer, and Spiros Michalakis for valuable input and inspiring discussions. We thank Emilio Onorati, Cambyse Rouzé, Daniel Stilck França, and James D. Watson for sharing a draft of their new results on efficiently predicting properties of states in thermal phases of matter with exponential decay of correlation and in quantum phases of matter with local topological quantum order[82]. LL is supported by Caltech Summer Undergraduate Research Fellowship (SURF), Barry M. Goldwater Scholarship, and Mellon Mays Undergraduate Fellowship. HH is supported by a Google PhD fellowship and a MediaTek Research Young Scholarship. JP acknowledges support from the U.S. Department of Energy Office of Science, Office of Advanced Scientific Computing Research (DE-NA0003525, DE-SC0020290), the U.S. Department of Energy, Office of Science, National Quantum Information Science Research Centers, Quantum Systems Accelerator, and the National Science Foundation (PHY-1733907). The Institute for Quantum Information and Matter is an NSF Physics Frontiers Center.

## Author contributions

H.H. and J.P. conceived the project. L.L. and H.H. developed the mathematical aspects of this work. L.L., H.H., S.L., and V.T. conducted the numerical experiments and wrote the open-source code. L.L., H.H., R.K., and J.P. wrote the paper.

## Competing interests

The authors declare no competing interests.
