## [Peer Review File · Nature Communications]

REVIEWER COMMENTS

Reviewer #1 (Remarks to the Author):

The paper introduces a classical machine algorithm that can efficiently predict ground state properties of an n -qubit gapped local Hamiltonian, using only $O(\log n)$ data about other Hamiltonians in the same state of matter. This represents an exponential improvement over previous results, which required a polynomial amount of data. The proposed approach also offers the advantage of scaling with $O(n \log n)$ training and prediction time. To overcome a lower bound found in previous work, the authors assume that the geometry of the n -qubit system is known. Here, the key idea behind the proposed algorithm is a nonlinear feature map with a geometric inductive bias built into the mapping. The algorithm learns the function by training an l_1 -regularized regression in the feature space.

The results of this paper are significant and noteworthy and goes beyond previous results (in particular, those in Ref 29) in several ways. First, as discussed above, the proposed algorithm gives an exponential improvement in the number of qubits for the sample complexity compared to Ref 29. Second, the proposed algorithm works for any probability distribution over x (the vector that parameterizes an n -qubit gapped geometrically local Hamiltonian), whereas the algorithm in Ref [29] works for only the uniform distribution. Third, the proposed algorithm gives an exponential-to-quasipolynomial improvement in the reciprocal of the prediction error for the sample complexity compared to [29].

The results of this paper are timely. Predicting the ground state of a quantum many-body system is indeed an important problem with far-reaching consequences for many fields, including physics, materials science, and chemistry.

I found the results of this paper to be presented clearly, with definitions, lemmas, and theorems stated clearly. The methodology is sound and I have not been able to find any technical errors in the paper. Additionally, the source code used to generate the numerical data in this paper is up on Github. There should be enough detail provided in the methods for the work to be reproduced.

There are some improvements that could be made though. Here are some suggestions/comments:

- Lemma 13 (in Appendix) writes that "[ρ] is a mixed state, i.e., it is positive semidefinite and has unit trace". But the latter is the definition of a "density operator", and not of a mixed state (which commonly refers to states that are not pure states).

- In Lemma 9 (in Appendix), the authors write that "log denotes the natural logarithm". But perhaps this should also be stated earlier? In the proof of Lemma 6, whether the inequality " $\log z \leq 2 \sqrt{z} - 1$ for $z > 0$ " holds depends on the base of the logarithm.

- While it might be commonly believed that the statement that "NP-complete problems can be solved in randomized polynomial time (Proposition 1)" is untrue, it might be helpful to state explicitly that this is the case. One needs to first assume that the statement above in quotes is false before the statement of "hardness" in the preceding paragraph saying "the problem of estimating ground state properties for the class of parameterized Hamiltonians ... considered in this work is hard for non-ML algorithms that cannot learn from data" follows.

- The following has a typo: "This is satisfied is $\log x > 2$ instead" [proof of Lemma 11]

- Ref 76 is missing the arXiv reference number.

Reviewer #2 (Remarks to the Author):

In this work, the authors present a novel machine-learning (ML) algorithm to predict few-body correlation functions in the ground-state of spatially local Hamiltonians. While their study builds upon the previous work [29], I believe the results reported here are a very significant step forward with respect to the state of the art. In addition, the draft is very well written (although the technical part could benefit from clarifications, see below). Finally, the topic is certainly very timely and of broad interest (even beyond quantum many-body physics). Because of these reasons, I believe this work is suitable for publication in Nature Communications, as I elaborate below.

In a previous work [29], a subset of the authors put forward a ML algorithm to study ground-state properties of quantum systems. The method required a polynomial amount of training data and computational time, and (based on some standard assumptions in complexity theory) was shown to be computationally more efficient than any other possible classical algorithm to solve the same problem. The algorithm presented in this work presents a striking improvement, requiring only an

amount of training data growing logarithmically in the number of the system qubits. In addition, the method also improves exponentially the scaling in the target accuracy.

These improvements are made possible by explicitly considering the geometric structure of the physical Hamiltonian under study. This ingredient was not considered before and makes the submitted work original not just from the merely technical point of view, as non-trivial physical input is used to obtain a better algorithm.

The most impressive part of the work pertains to the possibility of giving rigorous mathematical estimates on the efficiency of the method. However, the authors also report convincing numerical data supporting the mathematical claims. In light of these results, I think it is very likely that this approach will be used in practice in future works, paving the way to further improvements and progress along this line of research.

As the only piece of criticism, I think the technical part could be made clearer. In particular, the discussion is a bit hard to follow around Eq. II.1. For example: I didn't understand how $h_{\{j(c)\}}$ is defined from the definition given in the main text. It is written that $h_{\{j(c)\}}$ "is the few-body interaction term in the n -qubit Hamiltonian $H(x)$ that is parameterized by the variable x_c ". But aren't the terms h_j parameterized by vectors and not variables? Also the subsequent portion of text is not very clear.

I appreciated the explicit examples given in the supplemental material to explain the definitions. Perhaps, the authors could add further explanations to exemplify the definition (II.1). It would also be useful to mention in the main text that examples are given in the supplemental material.

In passing, I also found a trivial typo in the caption of Fig. 1: the point before ". The algorithm" should be replaced by a comma.

In summary, after the authors have considered my comments about the clarify of the technical part, I recommend publication.

Reviewer #3 (Remarks to the Author):

The authors describe a machine-learning model that can be trained via supervised learning to predict ground-state properties of rather general quantum spin models. The main focus is on providing rigorous guarantees on the required number of training samples, depending on the system sizes and on the target accuracy, and on demonstrating the computational complexity of training and inference. The model and algorithm presented here outperform in terms of sample and training complexity the previously known algorithms, introduced by the same research group in Ref. [29]. Specifically, the new model displays a logarithmic sample complexity in terms of system sizes, and a quasi-polynomial complexity as a function of the inverse of the prediction error. This improvement is due to addressing geometrically-local Hamiltonians and is achieved by introducing an inductive bias encoding geometric locality. The theory is supported by some numerical experiments on 2D antiferromagnetic Heisenberg models and the empirical training speed is consistent with the proposed theory.

The study of machine-learning algorithms that learn from data to solve otherwise intractable quantum many-body problems is a flourishing research line. The models presented in this manuscript represent a substantial step further compared to the earlier literature, in particular Ref.[29], by some of the same authors. These results support the perspective of using quantum devices to produce training data, so that computational tasks that are intractable for algorithms that did not learn from data can be solved via classical machine-learning algorithms. The results are sound, and the manuscript is, in my opinion, suitable for publication in Nature Communications. Still, some clarifications and some further details should be provided before publication. Below I report a list of issues and clarification requests.

Issues/clarifications:

-) After eq. 1.2, the authors state that $\langle y_I \rangle$ is approximately equal to the ground-state expectation value. I suggest providing already here a more precise definition or at least an explanation of what will be detailed later. How accurate should the approximation be? Also, it is not clear which results would remain unchanged if the training set included exact target values.

-) Five lines after eq. 1.2, the authors report the complexity lower bound of Ref. [29]. However, the meaning of the symbol Ω in the exponent is not clear.

-) Right before eq. 1.3, the authors define ϵ using the symbol Θ . The meaning of the latter is not clear.

-) Right before the beginning of Section II, the authors mention the combination with the classical shadow formalism. Also, at the end of Theorem 1, the authors mention that \hat{y}_l could be obtained by estimating expectation values via a given number of measurements, or through the classical shadow formalism. In general, the role of the classical shadow formalism in this study is not clear. The formal derivation uses, as target values, approximations for the expectation values with a fixed maximum discrepancy. Then, the numerical experiments pass through the classical shadow. It is not clear whether this is just for numerical convenience. Why not training the algorithm to directly predict the (approximation of) the expectation value, rather than the classical shadow?

-) In theorem 1, after eq. II.6, the authors state that the discrepancy between target values \hat{y}_l and expectation values is bounded by ϵ . The same variable ϵ bounds the mean squared error over the distribution D . Is it obvious to me that the same variable ϵ must occur in both places? In one case ϵ represents a maximum discrepancy, in the second an average value. I suggest providing an explanation.

-) In proposition 1, the authors mention that they consider a 2D gapped Hamiltonian. It is not clear to me why here one cannot consider a generic dimensionality.

-) In the numerical experiment, the authors adopt random, but strictly nonnegative, couplings. Being the square lattice bipartite, I guess this allows avoiding frustration effects. Is this what led the authors to the choice of nonnegative couplings?

-) In the numerical experiment, the authors adopt Fourier features instead of the indicator function. I suggest including a comment to explain if and why they expect their derived rigorous bounds to apply also to the LASSO model with this modification, and, eventually, to other more generic models.

-) In figure 2.A, the authors analyze the scaling of the average prediction error. However, it is not clear if the accuracy they reach is sufficiently high to study the physics of random Heisenberg models. What would be a target accuracy, for the quantities they consider, to perform practically useful calculations? Otherwise, the authors might at least present some relative-error data. This would help the reader gauging the performance.

-) I suggest reporting more details on the size of the training/validation/test sets also in the main text. These details are reported in a slightly opaque manner only at the end of the SM.

-) The analysis in fig. 2.A, central panel, is limited to $N=90$. Is it computationally challenging to produce much more training data with the adopted DMRG algorithm? In fact, the blue curve in this panel hints at a saturation effect, and it would be interesting (if feasible) to analyze the scaling for much larger N .

-) While the focus in this article is on rigorous complexity bounds, several recent articles empirically have explored the deep-learning of ground-state properties, beyond Refs. [65,66]. For example, in Refs. Phys. Rev. A 96, 042113 (2017) and Phys. Rev. E 102, 033301 (2020), ground-state energies of disordered quantum systems have been (deep) learned, and the learning speed has been empirically analyzed. In Ref. J. Phys. Chem. A 127, 339–355 (2023), forces from QMC have been machine learned. A large body of research focused on the database of molecules presented in Phys. Rev. Lett. 108, 058301 (2012). For example, in J. Chem. Theory Comput. 13, 5255-5264 (2017), DFT properties have been learning, reaching a level comparable to chemical accuracy.

Dear Reviewers,

We really appreciate your helpful comments in improving our work for the broad readership of *Nature Communications*. In the resubmission, we have addressed your comments and implemented corresponding changes in the manuscript. The changes in the manuscript are highlighted in blue. Thank you for your time in considering our work.

Sincerely on behalf of all authors,

Laura Lewis, Hsin-Yuan Huang, and John Preskill

Reviewer #1 (Remarks to the Author):

The paper introduces a classical machine algorithm that can efficiently predict ground state properties of an n -qubit gapped local Hamiltonian, using only $O(\log n)$ data about other Hamiltonians in the same state of matter. This represents an exponential improvement over previous results, which required a polynomial amount of data. The proposed approach also offers the advantage of scaling with $O(n \log n)$ training and prediction time. To overcome a lower bound found in previous work, the authors assume that the geometry of the n -qubit system is known. Here, the key idea behind the proposed algorithm is a nonlinear feature map with a geometric inductive bias built into the mapping. The algorithm learns the function by training an l_1 -regularized regression in the feature space.

The results of this paper are significant and noteworthy and goes beyond previous results (in particular, those in Ref 29) in several ways. First, as discussed above, the proposed algorithm gives an exponential improvement in the number of qubits for the sample complexity compared to Ref 29. Second, the proposed algorithm works for any probability distribution over x (the vector that parameterizes an n -qubit gapped geometrically local Hamiltonian), whereas the algorithm in Ref [29] works for only the uniform distribution. Third, the proposed algorithm gives an exponential-to-quasipolynomial improvement in the reciprocal of the prediction error for the sample complexity compared to [29].

The results of this paper are timely. Predicting the ground state of a quantum many-body system is indeed an important problem with far-reaching consequences for many fields,

including physics, materials science, and chemistry.

I found the results of this paper to be presented clearly, with definitions, lemmas, and theorems stated clearly. The methodology is sound and I have not been able to find any technical errors in the paper. Additionally, the source code used to generate the numerical data in this paper is up on Github. There should be enough detail provided in the methods for the work to be reproduced.

There are some improvements that could be made though. Here are some suggestions/comments:

Lemma 13 (in Appendix) writes that "[rho] is a mixed state, i.e., it is positive semidefinite and has unit trace". But the latter is the definition of a "density operator", and not of a mixed state (which commonly refers to states that are not pure states).

Author response: We thank the reviewer for their detailed reading of our work. We have made the suggested changes.

In Lemma 9 (in Appendix), the authors write that "log denotes the natural logarithm". But perhaps this should also be stated earlier? In the proof of Lemma 6, whether the inequality " $\log z \leq 2\sqrt{z} - 1$ for $z > 0$ " holds depends on the base of the logarithm.

Author response: Thank you for this suggestion. We have specified that log denotes the natural logarithm throughout Appendix A.

While it might be commonly believed that the statement that "NP-complete problems can be solved in randomized polynomial time (Proposition 1)" is untrue, it might be helpful to state explicitly that this is the case. One needs to first assume that the statement above in quotes is false before the statement of "hardness" in the preceding paragraph saying "the problem of estimating ground state properties for the class of parameterized Hamiltonians ... considered in this work is hard for non-ML algorithms that cannot learn from data" follows.

Author response: We thank the reviewer for this helpful comment, and we agree that this was unclear in the first draft. We have edited the manuscript accordingly to specify this assumption and the common belief that it holds true.

- The following has a typo: "This is satisfied is $\log x > 2$ instead" [proof of Lemma 11]
- Ref 76 is missing the arXiv reference number.

Author response: Thank you for spotting the typo and the missing reference number. We have fixed them in the revision.

Reviewer #2 (Remarks to the Author):

In this work, the authors present a novel machine-learning (ML) algorithm to predict few-body correlation functions in the ground-state of spatially local Hamiltonians. While their study builds upon the previous work [29], I believe the results reported here are a very significant step forward with respect to the state of the art. In addition, the draft is very well written (although

the technical part could benefit from clarifications, see below). Finally, the topic is certainly very timely and of broad interest (even beyond quantum many-body physics). Because of these reasons, I believe this work is suitable for publication in *Nature Communications*, as I elaborate below.

In a previous work [29], a subset of the authors put forward a ML algorithm to study ground-state properties of quantum systems. The method required a polynomial amount of training data and computational time, and (based on some standard assumptions in complexity theory) was shown to be computationally more efficient than any other possible classical algorithm to solve the same problem. The algorithm presented in this work presents a striking improvement, requiring only an amount of training data growing logarithmically in the number of the system qubits. In addition, the method also improves exponentially the scaling in the target accuracy.

These improvements are made possible by explicitly considering the geometric structure of the physical Hamiltonian under study. This ingredient was not considered before and makes the submitted work original not just from the merely technical point of view, as non-trivial physical input is used to obtain a better algorithm.

The most impressive part of the work pertains to the possibility of giving rigorous mathematical estimates on the efficiency of the method. However, the authors also report convincing numerical data supporting the mathematical claims. In light of these results, I think it is very likely that this approach will be used in practice in future works, paving the way to further improvements and progress along this line of research.

As the only piece of criticism, I think the technical part could be made clearer. In particular, the discussion is a bit hard to follow around Eq. III. For example: I didn't understand how $h_{j(c)}$ is defined from the definition given in the main text. It is written that $h_{j(c)}$ "is the few-body interaction term in the n -qubit Hamiltonian $H(x)$ that is parameterized by the variable x_c ". But aren't the terms h_j parameterized by vectors and not variables? Also the subsequent portion of text is not very clear.

Author response: We thank the reviewer for bringing this to our attention and apologize for the lack of clarity. As the reviewer states, each h_j is parameterized by a vector \vec{x}_j . What we mean by saying that " $h_{j(c)}$ is parameterized by the variable x_c " is that the vector \vec{x}_j that parameterizes h_j includes the c th entry of the vector x , which we denote by the variable x_c . For example, suppose $h_j = h_j(\vec{x}_j)$ for $\vec{x}_j = (x_1, x_2, x_4)$. Then, we would write $h_{j(2)}$ to denote that \vec{x}_j includes the variable x_2 . We have endeavored to clarify this in the manuscript along with the subsequent text.

I appreciated the explicit examples given in the supplemental material to explain the definitions. Perhaps, the authors could add further explanations to exemplify the definition (II.1). It would also be useful to mention in the main text that examples are given in the supplemental material.

Author response: We appreciate this feedback and have included a reference to the supplemental material. An example illustrating Eq. (II.1) is given implicitly in Fig. 3. We

have edited the manuscript to make this connection more explicit.

In passing, I also found a trivial typo in the caption of Fig. 1: the point before ". The algorithm" should be replaced by a comma.

Author response: The authors thank the reviewer for their detailed reading of our work. The change has been made.

In summary, after the authors have considered my comments about the clarify of the technical part, I recommend publication.

Reviewer #3 (Remarks to the Author):

The authors describe a machine-learning model that can be trained via supervised learning to predict ground-state properties of rather general quantum spin models. The main focus is on providing rigorous guarantees on the required number of training samples, depending on the system sizes and on the target accuracy, and on demonstrating the computational complexity of training and inference. The model and algorithm presented here outperform in terms of sample and training complexity the previously known algorithms, introduced by the same research group in [Huang et al., 2022]. Specifically, the new model displays a logarithmic sample complexity in terms of system sizes, and a quasi-polynomial complexity as a function of the inverse of the prediction error. This improvement is due to addressing geometrically-local Hamiltonians and is achieved by introducing an inductive bias encoding geometric locality. The theory is supported by some numerical experiments on 2D antiferromagnetic Heisenberg models and the empirical training speed is consistent with the proposed theory.

The study of machine-learning algorithms that learn from data to solve otherwise intractable quantum many-body problems is a flourishing research line. The models presented in this manuscript represent a substantial step further compared to the earlier literature, in particular Ref.[29], by some of the same authors. These results support the perspective of using quantum devices to produce training data, so that computational tasks that are intractable for algorithms that did not learn from data can be solved via classical machine-learning algorithms. The results are sound, and the manuscript is, in my opinion, suitable for publication in Nature Communications. Still, some clarifications and some further details should be provided before publication. Below I report a list of issues and clarification requests.

Issues/clarifications:

-) After eq. I.2, the authors state that y_l is approximately equal to the ground-state expectation value. I suggest providing already here a more precise definition or at least an explanation of what will be detailed later. How accurate should the approximation be? Also, it is not clear which results would remain unchanged if the training set included exact target values.

Author response: Thank you for this suggestion. We have added a definition of this approximation and clarified that the result still holds if the training set includes exact target values. In fact, the guarantees would only improve in this case. The relationship between the

amount of error in the data and the prediction error/sample complexity is detailed further in Section C.2 of the supplementary material.

-) *Five lines after eq. I.2, the authors report the complexity lower bound of Ref. [29]. However, the meaning of the symbol Ω in the exponent is not clear.*

-) *Right before eq. I.3, the authors define ϵ using the symbol Θ . The meaning of the latter is not clear.*

Author response: Thank you for your valuable feedback. We appreciate your attention to detail and would like to clarify the symbols Ω and Θ , which are used in mathematics to denote the asymptotic scaling of functions and are commonly used in the field of complexity analysis of algorithms. The symbol Ω (known as Big Omega notation) represents an asymptotic lower bound, indicating the minimum resources required by any algorithm to solve the problem. The more common symbol \mathcal{O} (known as Big O notation) represents an asymptotic upper bound, indicating the resource that the best known algorithm needs to solve the problem. The symbol Θ (known as Big Theta notation) represents a matching asymptotic upper and lower bound, hence exactly characterizes the resource needed for a problem. These notations are essential for comparing the efficiency of an algorithm to solve a problem in the amount of data and the computational time (or other resources). Given that general audiences are likely unfamiliar with these notations, we have clarified their meaning in the revised version.

-) *Right before the beginning of Section II, the authors mention the combination with the classical shadow formalism. Also, at the end of Theorem 1, the authors mention that y_ℓ could be obtained by estimating expectation values via a given number of measurements, or through the classical shadow formalism. In general, the role of the classical shadow formalism in this study is not clear. The formal derivation uses, as target values, approximations for the expectation values with a fixed maximum discrepancy. Then, the numerical experiments pass through the classical shadow. It is not clear whether this is just for numerical convenience. Why not training the algorithm to directly predict the (approximation of) the expectation value, rather than the classical shadow?*

Author response: We thank the reviewer for this feedback and have endeavored to make the role of the classical shadow more clear in the manuscript sections suggested. Overall, we can use the classical shadow formalism to generate training data to give as input to our ML algorithm. As the reviewer mentioned, one way is to use it to generate the training labels y_ℓ . However, the method we are mainly trying to emphasize is the following. If we instead give our machine learning algorithm training data consisting of parameters x_ℓ along with a classical shadow representation of the ground state (rather than the label y_ℓ that approximates the ground state property), then we can use a slight modification of the ML algorithm to predict ground state representations. This is what is referred to at the end of Section I as the reviewer referenced and is described in more detail in Corollary 1 and Appendix C.2. This is a more powerful technique than that of Theorem 1 alone, as generating a ground state representation allows for the prediction of many observables without the need to retrain the model. In contrast, in Theorem 1, we can only predict just one property. We hope that this explanation

clarifies the role of classical shadow for the reviewer and have added a brief summary of this in the main text.

In regards to the numerical experiments, the reason why we pass through the classical shadow is not for numerical convenience but to train an ML model that can produce ground state representations which can be used to predict many properties of the ground state. By training the algorithm to predict the ground state representation via classical shadows, we can then use this to predict ground state properties for a wide range of observables. This is in contrast to the case where we would only be able to predict a single property of the ground state, which is what occurs when we train the algorithm to directly predict the expectation value itself.

-) *In theorem 1, after eq. II.6, the authors state that the discrepancy between target values y_i and expectation values is bounded by ϵ . The same variable ϵ bounds the mean squared error over the distribution D . Is it obvious to me that the same variable ϵ must occur in both places? In one case ϵ represents a maximum discrepancy, in the second an average value. I suggest providing an explanation.*

Author response: Thank you for your question. It is not immediately obvious that the same variable ϵ should be in both places here. This is rigorously explained and proven in Section C.2 of the supplementary material.

-) *In proposition 1, the authors mention that they consider a 2D gapped Hamiltonian. It is not clear to me why here one cannot consider a generic dimensionality.*

Author response: Thank you for your comment and close reading of the manuscript. The key idea of this proposition is to say that even just for the case of 2D Hamiltonians and predicting properties corresponding to single-qubit observables, this problem is still hard. Because we can reduce Hamiltonians in higher dimensions to this 2D case (e.g., by restricting the high-dimensional Hamiltonian to be nontrivial only on a 2D slice of the high-dimensional lattice), the problem of predicting ground state properties for Hamiltonians in higher dimensions should be even harder. We have edited the manuscript to clarify this.

-) *In the numerical experiment, the authors adopt random, but strictly nonnegative, couplings. Being the square lattice bipartite, I guess this allows avoiding frustration effects. Is this what led the authors to the choice of nonnegative couplings?*

Author response: Thank you for this question. When the couplings are negative, minimizing the energy caused by the Hamiltonians encourages the qubits to align (ferromagnetic). Instead of avoiding frustration effects, we chose the nonnegative couplings to enforce frustration effects. This way, the problem of finding ground states is harder, and we want to emphasize that our ML model can still solve this more challenging problem.

Another reason we consider nonnegative couplings is because this is the same class of Hamiltonians considered in the previous paper ([Huang et al., 2022]). We wanted to compare our numerical results to theirs, so we considered the same class of Hamiltonians to have a fair comparison. We have edited the manuscript to reflect this.

-) In the numerical experiment, the authors adopt Fourier features instead of the indicator function. I suggest including a comment to explain if and why they expect their derived rigorous bounds to apply also to the LASSO model with this modification, and, eventually, to other more generic models.

Author response: We appreciate this suggestion and agree that this deserves more emphasis in the main text. We expect that the Fourier features will give similar rigorous bounds, but the proof will become much more complicated. The indicator function is mainly a tool used to make the proofs simpler but is often not used in practice. Meanwhile, the random Fourier features are much more commonly used, and similar arguments hold because Fourier features can approximate any function, similarly to how we used the indicator functions in the proof. The random Fourier feature is essentially a single layer of a randomly initialized neural network. We have added these points to the main text.

-) In figure 2.A, the authors analyze the scaling of the average prediction error. However, it is not clear if the accuracy they reach is sufficiently high to study the physics of random Heisenberg models. What would be a target accuracy, for the quantities they consider, to perform practically useful calculations? Otherwise, the authors might at least present some relative-error data. This would help the reader gauging the performance.

Author response: We agree that in the initial manuscript, it is not clear how to gauge the performance of our algorithm, and thank the reviewer for this suggestion. For this, we refer to Figure 3 in [Huang et al., 2022] (Ref. [29] in the original submission). This figure illustrates that for the average prediction error achieved in their work, the predictions by the ML algorithm match the exact simulated values very closely and are thus at a high enough accuracy to study the physics of random Heisenberg models.

In this work, we emphasize that because we can use a significantly smaller amount of training data to achieve the same prediction error as [Huang et al., 2022], this means that significantly less data is needed to perform these practically useful calculations. We have added a similar explanation to the manuscript.

-) I suggest reporting more details on the size of the training/validation/test sets also in the main text. These details are reported in a slightly opaque manner only at the end of the SM.

Author response: We thank the reviewer for this feedback. We have added these details to the main text and clarified this in the supplemental material.

-) The analysis in fig. 2.A, central panel, is limited to $N = 90$. Is it computationally challenging to produce much more training data with the adopted DMRG algorithm? In fact, the blue curve in this panel hints at a saturation effect, and it would be interesting (if feasible) to analyze the scaling for much larger N .

Author response: Thank you for the question. It is indeed computationally challenging to produce large training data due to the intensive calculations used by the DMRG algorithm. The algorithm works well for 1D systems, but for 2D systems, DMRG is an exponential-time algorithm. After obtaining bigger training data from DMRG, we have updated the plot to

include larger training data sizes (up to $N = 450$).

Log-log plots are given to showcase the dependence of the average prediction error ϵ on the measurement per system (T) and the training set size (N). The plot shows that the error $\epsilon = \epsilon(N)$ continues to decrease as N increases. For our new algorithm, the plot for large training data sizes $N \geq 50$ is approximately linear with a slope α of around $1/3.5$, which implies that $\epsilon = C/N^{1/3.5}$. The seemingly decreasing slope at around $N = 90$ in the original plot is a visual artifact that arises from the polynomial scaling of ϵ on N in an $\epsilon - N$ plot, which is removed from a $\log(\epsilon) - \log(N)$ plot. The result is shown in Fig. 2A.

-) While the focus in this article is on rigorous complexity bounds, several recent articles empirically have explored the deep-learning of ground-state properties, beyond Refs. [65,66]. For example, in Refs. *Phys. Rev. A* 96, 042113 (2017) and *Phys. Rev. E* 102, 033301 (2020), ground-state energies of disordered quantum systems have been (deep) learned, and the learning speed has been empirically analyzed. In Ref. *J. Phys. Chem. A* 127, 339–355 (2023), forces from QMC have been machine learned. A large body of research focused on the database of molecules presented in *Phys. Rev. Lett.* 108, 058301 (2012). For example, in *J. Chem. Theory Comput.* 13, 5255-5264 (2017), DFT properties have been learned, reaching a level comparable to chemical accuracy.

Author response: We thank the reviewer for bringing these references to our attention and have added them to the introduction.

References

[Huang et al., 2022] Huang, H.-Y., Kueng, R., Torlai, G., Albert, V. V., and Preskill, J. (2022). Provably efficient machine learning for quantum many-body problems. *Science*, 377(6613):eabk3333.

REVIEWERS' COMMENTS

Reviewer #1 (Remarks to the Author):

The authors have addressed all my comments and have implemented all the suggestions in my previous review.

The results presented are timely and important. The presentation in this new version is clear and the results are stated clearly. Additionally, I believe that this work stands to be of interest to the readership of Nature Communications. For these reasons, I recommend the publication of this manuscript.

Reviewer #2 (Remarks to the Author):

The authors have fully addressed my comments. I therefore recommend publication of the manuscript.

Reviewer #3 (Remarks to the Author):

The authors have exhaustively addressed the comments I made in my previous report on the original version of the manuscript. In particular, I notice the increased size in the numerical experiments, which allow to better analyze the scaling of the prediction error. Also, the role of the classical shadows is better described in the revised manuscript.

I recommend publication of the manuscript in Nature Communication.

As a final minor remark, I suggest the authors to carefully analyze the comments of the possible role of frustration in the 2D Heisenberg model they address. If I correctly understood the notation, the authors consider only non negative couplings on a bipartite lattice. Despite of the randomness, it is not obvious to me that this model should display frustration effects, since an antiferromagnetic ordering should satisfy all interaction terms. Usually, the term frustration is invoked when also diagonal couplings are present, or on different lattices.

Dear Reviewers,

We really appreciate your helpful comments in improving our work for the broad readership of *Nature Communications*. In the resubmission, we have addressed your remaining comments and implemented corresponding changes in the manuscript. The changes in the manuscript are highlighted in blue. Thank you for your time in considering our work.

Sincerely on behalf of all authors,

Laura Lewis, Hsin-Yuan Huang, and John Preskill

Reviewer #1 (Remarks to the Author):

The authors have addressed all my comments and have implemented all the suggestions in my previous review.

The results presented are timely and important. The presentation in this new version is clear and the results are stated clearly. Additionally, I believe that this work stands to be of interest to the readership of Nature Communications. For these reasons, I recommend the publication of this manuscript.

Author response: We thank the reviewer for their positive comments and support.

Reviewer #2 (Remarks to the Author):

The authors have fully addressed my comments. I therefore recommend publication of the manuscript.

Author response: We thank the reviewer for their positive comments and support.

Reviewer #3 (Remarks to the Author):

The authors have exhaustively addressed the comments I made in my previous report on the original version of the manuscript. In particular, I notice the increased size in the numerical experiments, which allow to better analyze the scaling of the prediction error. Also, the role of the classical shadows is better described in the revised manuscript. I recommend publication of the manuscript in Nature Communication.

Author response: We thank the reviewer for their positive comments and support.

As a final minor remark, I suggest the authors to carefully analyze the comments of the possible role of frustration in the 2D Heisenberg model they address. If I correctly understood the notation, the authors consider only non negative couplings on a bipartite lattice. Despite of the randomness, it is not obvious to me that this model should display frustration effects, since an antiferromagnetic ordering should satisfy all interaction terms. Usually, the term frustration is invoked when also diagonal couplings are present, or on different lattices.

Author response: We thank the reviewer for raising this remark. We have provided more detailed explanation on the possible role of frustration in this model.